# Activation of Chaperone-Mediated Autophagy Inhibits the Aryl Hydrocarbon Receptor Function by Degrading This Receptor in Human Lung Epithelial Carcinoma A549 Cells

**DOI:** 10.3390/ijms242015116

**Published:** 2023-10-12

**Authors:** Rui Xiong, Dan Shao, Sandra Do, William K. Chan

**Affiliations:** Department of Pharmaceutics & Medicinal Chemistry, Thomas J. Long School of Pharmacy, University of the Pacific, Stockton, CA 95211, USA; r_xiong1@u.pacific.edu (R.X.); d_shao@u.pacific.edu (D.S.); s_do10@u.pacific.edu (S.D.)

**Keywords:** aryl hydrocarbon receptor, chaperone-mediated autophagy, A549, 6-AN, LAMP2, CQ

## Abstract

The aryl hydrocarbon receptor (AHR) is a ligand-activated transcription factor and a substrate protein of a Cullin 4B E3 ligase complex responsible for diverse cellular processes. In the lung, this receptor is responsible for the bioactivation of benzo[a]pyrene during tumorigenesis. Realizing that the AHR function is affected by its expression level, we are interested in the degradation mechanism of AHR in the lung. Here, we have investigated the mechanism responsible for AHR degradation using human lung epithelial A549 cells. We have observed that the AHR protein levels increase in the presence of chloroquine (CQ), an autophagy inhibitor, in a dose-dependent manner. Treatment with 6-aminonicotinamide (6-AN), a chaperone-mediated autophagy (CMA) activator, decreases AHR protein levels in a concentration-dependent and time-dependent manner. This decrease suppresses the ligand-dependent activation of the AHR target gene transcription, and can be reversed by CQ but not MG132. Knockdown of lysosome-associated membrane protein 2 (LAMP2), but not autophagy-related 5 (ATG5), suppresses the chloroquine-mediated increase in the AHR protein. AHR is resistant to CMA when its CMA motif is mutated. Suppression of the epithelial-to-mesenchymal transition in A549 cells is observed when the *AHR* gene is knocked out or the AHR protein level is reduced by 6-AN. Collectively, we have provided evidence supporting that AHR is continuously undergoing CMA and activation of CMA suppresses the AHR function in A549 cells.

## 1. Introduction

The aryl hydrocarbon receptor (AHR) is a biological sensor that alters gene expression in response to many environmental pollutants (such as dioxins and polychlorinated biphenyl compounds) and flavonoids [1,2]. Many tryptophan metabolites (such as 6-formylindolo (3,2-b) carbazole (FICZ)) have been shown to be the endogenous ligands of this receptor. These metabolites can be generated in the gut microbiome and subsequently activate AHR in human cells [3]. AHR suppresses the immune response in part by promoting naïve T cell differentiation into T regulatory cells [4]. It also drives the growth of many tumors [5], promotes insulin resistance [6], and is a drug target for psoriasis treatment [7]. Interestingly, AHR has been implicated as a drug target for the treatment of SARS-CoV2 infection since AHR may alter lung function in favor of supporting SARS-CoV2 infection [8,9]. Regarding the role of AHR in lung tumorigenesis, there are conflicting reports on the effect of AHR on the epithelial–mesenchymal transition (EMT). Although some researchers observed a positive correlation between the AHR action and the aggressive phenotypes of invasion and metastasis in lung and other cell types [10,11,12,13], others reported that AHR suppressed metastasis by inhibiting lung EMT [14,15]. Nonetheless, the expression levels of AHR must correlate positively with its function in the lung. We are interested in studying the degradation mechanisms of AHR in lung epithelial cells in affecting the AHR function.

AHR exists as a complex in the cytoplasm with heat shock protein 90 (HSP90), co-chaperone protein p23, hepatitis B virus-x associated protein 2 (XAP2), and possibly proto-oncogene tyrosine-protein kinase (Src) [16,17,18]. Exposure of the nuclear localization sequence of AHR after ligand binding leads to nuclear translocation of the complex. The binding of the aryl hydrocarbon receptor nuclear translocator (ARNT) to AHR in the nucleus dissociates the complex [18]. The AHR–ARNT heterodimer binds to its dioxin response element (DRE) enhancer, activating the transcription of its target genes, such as cytochrome P450 1a1 (*CYP1A1*). Alternatively, AHR serves as a substrate protein that recruits its target proteins (for example, ERα) to CUL4B E3 ligase for proteasomal degradation [19].

Interestingly, AHR is degraded via autophagy in many human cell lines, namely lung A549, liver Hep3B, and breast T-47D and triple-negative MDA-MB-468 cells [20]. Proteins can be selectively degraded via selective macroautophagy and chaperone-mediated autophagy (CMA) [21,22]. Selective macroautophagy employs p62 to escort client proteins into autophagosomes by interaction with microtubule-associated protein 1 light chain (LC3) at the autophagosome membrane, followed by fusion with lysosomes. This fusion leads to lysosomal degradation of the client proteins. Many autophagy-related gene proteins, such as ATG5, are essential for LC3 lipidation at the membrane during the process of selective macroautophagy [23,24]. Alternatively, client proteins are escorted by heat shock cognate 70 kDa protein (HSC70) to lysosomes by interaction with the lysosomal membrane glycoprotein LAMP2A, leading to internalization of the client proteins for degradation. This process is called chaperone-mediated autophagy (CMA). LAMP2A plays an important role in the CMA process by mediating the lysosomal degradation of proteins in response to various stresses and keeping the normal turnover of proteins with a long biological half-life [25]. We have observed that degradation of AHR via autophagy is cell-line specific: AHR undergoes selective macroautophagy in human cervical HeLa cells [20] and CMA in triple-negative breast cancer MDA-MB-468 cells [26]. Here, we provide evidence that CMA is responsible for AHR degradation in human lung epithelial A549 cells. Targeting the AHR degradation mechanism can be a viable approach in the lung since modulation of the AHR protein level via CMA alters the AHR function in A549 cells.

## 2. Results

### 2.1. Subsection

#### 2.1.1. CQ Increases the AHR Protein Levels of A549 Cells in a Functionally Relevant Manner

CQ is a general autophagy inhibitor that inhibits the lysosomal proteases. We addressed whether AHR undergoes autophagy in A549 cells by examining AHR contents in the presence of CQ. Treatment of A549 cells with 60 and 100 μM CQ for 6 h increased the AHR protein levels to 1.5- and 2-fold, respectively (Figure 1A). To address whether a 2-fold increase in the AHR content would elicit any functional significance, we examined the effect of this fold change on the ligand-dependent upregulation of the AHR target gene *CYP1A1*. As expected, treatment with 100 μM CQ for 6 h increased the AHR protein content by 2-fold in the presence or absence of benzo[α]pyrene (BaP) (Figure 1B). Treatment with 5 μM BaP alone for 4 h reduced the AHR protein content due to proteasomal degradation of AHR after ligand binding [27,28]. We observed that BaP caused a 10-fold increase in the *CYP1A1* transcript after 4 h of treatment (Figure 1C). This upregulation was enhanced from 10- to 38-fold in the presence of 100 μM CQ, showing that a 2-fold increase in the AHR content (when compared between BaP and BaP/CQ treatments) caused a significant increase in the AHR function. Treatment with CQ alone did not change the *CYP1A1* transcript level, confirming that the upregulation of the *CYP1A1* gene transcription was mediated by BaP, not CQ. Collectively, we conclude that a 2-fold increase in the content of A549 AHR by CQ can significantly enhance the BaP-dependent AHR function.

#### 2.1.2. 6-AN Reduces the AHR Protein Content of A549 Cells in a Dose- and Time-Dependent Manner with Functional Relevance

Next, we used activators of either selective macroautophagy or CMA to explore the autophagy mechanism for AHR degradation in A549 cells. Treatment with 6-AN, a CMA activator, caused a dose-dependent reduction in the AHR protein content (Figure 2A). This reduction was time-dependent as well, supporting that activation of CMA can effectively reduce the AHR protein content in A549 cells (Figure 2B). When compared to the reduction of the AHR protein content upon ligand (BaP) treatment, both 6-AN and BaP reduced the AHR protein content to a similar extent after 24 h of treatment. This prolonged suppression of the AHR levels is not surprising since TCDD and 3MC can cause a similar suppression of AHR for 24 h in Hepa1c1c7 cells [29]. However, unlike the BaP-induced AHR proteasomal degradation, the reduction of the AHR protein content by 6-AN was reversed in the presence of an autophagy inhibitor (CQ) but not a proteasomal inhibitor (MG132) (Figure 2C,D). Treatment with MG132 even suppressed the AHR protein content further. This is consistent with our finding that treatment of A549 cells with MG132 alone for 6 h also reduced the AHR protein levels, and this reduction was reversed in the presence of CQ (Figure 2E), which can be explained by the literature report that MG132 can induce autophagy [30,31]. In fact, lactacystin, another proteasomal degradation inhibitor, has been reported to activate autophagy partly by upregulating LC3 expression [32]. Interestingly, MG132 reversed the inhibition of AHR autophagy by CQ (Figure 2E), consistent with the notion that AHR undergoes autophagy and MG132 can activate autophagy to degrade AHR. However, both metformin (Met) and rapamycin (Rap), which are selective macroautophagy activators, did not reduce but increased the AHR protein content (Figure 2F,G). This increase can be explained by the crosstalk between selective macroautophagy and CMA in that activation of one autophagy mechanism may negatively regulate the other [33]. In any case, selective macroautophagy does not seem to be involved in the degradation of AHR in A549 cells. To address whether 6-AN can suppress AHR function, we examined the effect of 6-AN on the ligand-induced, AHR-dependent activation of the *CYP1A1* gene transcription. We observed that both BaP (5 μM) and FICZ (1 μM) upregulated the *CYP1A1* message by 10- and 20-fold, respectively, after 6 h of treatment (Figure 2H). 6-AN (100 μM) effectively suppressed the induction to less than 5-fold in both cases, showing that the reduction in the AHR protein content by 6-AN to about 40% of its content in A549 cells significantly hampered its function.

#### 2.1.3. Knockdown of LAMP2 in A549 Cells Abolishes the CQ-Mediated Increase of the AHR Protein Content

LAMP2A, a lysosomal membrane-bound protein, is responsible for the internalization of CMA substrates into lysosomes for degradation. To further address whether CMA could degrade AHR in A549 cells, we used shRNA to knock down the *LAMP2* messages, which include the *LAMP2A* message, in A549 cells to see whether the AHR protein content is affected when CMA is less active. Using LAMP2-specific shRNA/siRNA to knock down the *LAMP2A* message is a common approach to downregulate LAMP2A expression. We observed that the AHR protein content increased to 1.7-fold when LAMP2 was down-regulated (Figure 3A), suggesting that LAMP2A plays a role in the degradation of AHR. Results from the LAMP2 Western showed a stretch of the LAMP2 region that represented both LAMP2A and LAMP2B. Additionally, CQ was unable to increase the AHR protein content when LAMP2 was downregulated, supporting that AHR is degraded via CMA in A549 cells. In contrast, knocking down ATG5, which is necessary for LC3 lipidation into LC3-phosphatidylethanolamine conjugate (LC3-II) during the formation of autophagosomes, decreased the AHR protein levels to 76% when compared to the wild-type A549 cells, suggesting that ATG5 is not involved in the degradation of AHR (Figure 3B). The reduction in the AHR content could be the compensatory mechanism that might activate CMA, since this kind of crosstalk between selective macroautophagy and CMA has been reported in the literature [33]. Treatment with CQ did not change the extent of the AHR protein content increase between the wild-type and ATG5 knockdown cells, suggesting that selective macroautophagy is not involved in AHR degradation in A549 cells.

#### 2.1.4. Degradation of AHR via CMA Is Dependent on the NEKFF Motif of AHR in A549 Cells

Earlier, we reported that NEKFF at amino acids 558–562 of the human AHR could be the CMA motif that allows interaction with HSC70, followed by recruitment to LAMP2A for degradation [26]. To further investigate the CMA-mediated AHR degradation in A549 cells, we examined the necessity of NEKFF for AHR degradation. We had previously generated a few GFP fusions of the wild-type human AHR and its mutants by altering the NEKFF sequence via site-directed mutagenesis [26]. We confirmed the sequence by sequencing the whole AHR cDNAs of the GFP-AHR (NEKFF) and the GFP-AHR mutant (NAKAF). Although the only differences in the amino acid sequence between the GFP-AHR (558-NEKFF-562) and GFP-AHR mutant (558-NAKAF-562) were E559A and F561A, we noticed two random mutations resulting in I581T and S590N when compared to the published human AHR cDNA sequence (NM_001621). To minimize any interference of the A549 AHR on the degradation of the GFP-AHR and GFP-AHR mutant due to shared machinery, we transfected the GFP plasmid into the *AHR* knockout A549 cells. First, we generated five clones of *AHR* knockout A549 cells using the clustered regularly interspaced short palindromic repeats/CRISPR-associated protein 9 (CRISPR/Cas9) strategy, namely 4H2, 2F6, 3C9, 5F2, and 5G11 (Figure 3C). Using the Synthego ICE v3.0 knockout online analysis software, all clones were 100% edited with insertions or deletions (% indel score), and clone 5G11 was determined to be a homologous bi-allelic knockout with 47 nucleotide deletions at exon 2, resulting in a frameshift mutation. Both wild-type and mutant *ahr* cDNAs were cloned downstream to the GFP cDNA, followed by transient transfection into the *AHR* knockout A549 5G11 cells. The GFP fusion of the wild-type AHR containing NEKFF showed similar suppression as the A549 AHR when treated with 100 μM 6-AN for 24 h (Figure 3D vs. Figure 2A). However, the GFP fusion of the NAKAF mutant was resistant to degradation upon 6-AN treatment, supporting that NEKFF is the CMA motif and AHR undergoes the CMA-mediated degradation. This observation is consistent with our immunoprecipitation results that treatment with CQ enhanced the interactions of AHR with HSC70 and LAMP2 in vitro (Figure 3E). These interactions were also observed in MDA-MB-468 cells [26].

#### 2.1.5. Autophagy of AHR Is Ongoing in the Background While AHR Is Undergoing Rapid Degradation via the Ubiquitin–Proteasome System after Treatment with an AHR Ligand in A549 Cells

It is well accepted that upon ligand treatment, AHR undergoes proteasomal degradation within hours of treatment [28]. We were interested in how autophagy of AHR is affected by ligand treatment in A549 cells. As expected, AHR was degraded to 40% of its wild-type content within 2 h of 5 μM BaP treatment via 26S proteasome since cotreatment with 10 μM MG132 completely reversed the degradation (Figure 4A). Cotreatment of BaP-treated cells with 100 μM CQ, however, increased the AHR protein levels to 1.8-fold when compared to cells treated with BaP alone. This fold change was close to the increase we observed when we treated A549 cells with 100 μM CQ for 6 h (Figure 1A), suggesting that AHR underwent the usual rate of degradation via autophagy while proteasomal degradation of AHR was triggered by a ligand. Interestingly, cotreatment of A549 cells with 5 μM BaP and 10 μM MG132 for an additional 4 h (a total of 6 h) showed that the AHR protein levels pronouncedly dropped to less than 30% content (Figure 4A,B). Considering that MG132 can activate autophagy, we examined the possibility that AHR underwent autophagy between the second and sixth hours of MG132 treatment. We exposed the BaP-treated cells to both MG132 and CQ for 6 h and observed only a 2-fold increase of the AHR protein levels (from 29% to 58%) (unpublished data). The strong reduction in the AHR protein levels from the second to the sixth hours of MG132 treatment cannot be fully explained by merely the lysosomal degradation of AHR. The AHR protein levels were similar from 6 h up to 24 h after BaP treatment (Figure 2B, right). Interestingly, similar inhibition of AHR autophagy by CQ was observed within the 6 h to 24 h period since a 1.5 to 1.8-fold increase in the AHR levels was observed when we compared BaP and BaP/CQ treatment groups (Figure 4A–D). Figure 4E contains the representative Western blot images of Figure 4A–D, which showed that from 6 h to 24 h, the AHR levels increased in the presence of CQ, whereas MG132 did not have any effect. Collectively, we concluded that autophagy is likely involved in maintaining AHR levels after ligand treatment.

#### 2.1.6. Autophagy Is Not Involved in the Quick-Onset Degradation of AHR Triggered by a Low Dose of Geldanamycin (GA) But Is Involved in Controlling the AHR Levels after Both Low and High Doses of GA Treatment in A549 Cells

It is known that treatment with a low dose of GA (0.1 μM) causes proteasomal degradation of AHR [34]. We examined whether autophagy might also be involved in this GA-mediated degradation. Our results showed that treatment with a low dose of GA (0.1 μM) for 2 h caused a pronounced reduction in the AHR content to 17% in A549 cells, which could be partially reversed to 41% content by 10 μM MG132 (Figure 5A). The fact that MG132 could reverse 100% of the BaP-mediated degradation of AHR (Figure 4A) and, to a much lesser extent, the GA-mediated degradation of AHR (Figure 5A), suggested that the GA effect on AHR degradation is more complex than merely proteasomal degradation. On the other hand, inhibition of autophagy by CQ (100 μM) did not alter the low AHR levels caused by 0.1 μM GA. Realizing that a low dose of GA (0.5 μM) could inhibit autophagy by suppressing the Atg7, Beclin-1, and ULK1 protein levels, as reflected by a reduction in the autophagic flux [35], AHR degradation via autophagy might have been inhibited in the presence of 0.1 μM GA. Although two additional hours of GA exposure (which was 4 h treatment) did not further suppress AHR but rather caused a slight increase in content (28% vs. 17%), MG132 restored more AHR content when compared to 2 h (66% vs. 41%). These results suggested that there might be newly synthesized AHR between 2 and 4 h that was degraded by proteasomal degradation (Figure 5B). In addition, CQ also increased the AHR content from 28% to 43%, suggesting autophagy was also at play between the second and fourth hours of GA treatment.

It has been reported that treatment with a high dose of GA (2–10 μM) for 24 h causes degradation of IKK via the ATG5-dependent autophagy in 293, HeLa, and B and T cells [36]. The activity of CMA is doubled when IMR-90 cells are treated with 2 μM of GA [37]. Therefore, we examined whether autophagy and proteasomal degradation might play a role in the sustained low levels of AHR many hours after a high dose of GA treatment. We treated the cells with 1 μM GA (high dose) for 24 h, followed by exposure to CQ, MG132, or the DMSO vehicle in the last 6 h before harvesting in A549 cells. We used 1 μM because a higher dose (2 μM) of GA caused apparent cell death after 24 h. We observed that CQ increased the AHR levels significantly from 29 to 58%, whereas MG132 did not alter the AHR levels in a statistically significant manner (Figure 5C,D), showing that autophagy, but not proteasomal degradation, is essential for maintaining the AHR protein levels. The slight reduction in AHR levels by MG132 in this case is probably due to the activation of autophagy by MG132, as demonstrated elsewhere [30,31]. Results from the time-dependent experiment showed that a high dose of GA suppressed the AHR protein levels more effectively than a low dose of GA (Figure 5E), supporting that a high dose of GA causes more AHR degradation, possibly through autophagy. However, we cannot rule out the possibility that a higher dose of GA may also cause an increase in the proteasomal degradation of AHR. Collectively, AHR undergoes autophagy, which can be triggered by a high dose of GA and blocked by CQ (Figure 5C,E).

#### 2.1.7. 6-AN Is Not an AHR Ligand

Although our data supported that 6-AN activates CMA to degrade AHR, we examined whether 6-AN is also an AHR ligand, knowing that the binding of an AHR ligand normally causes AHR protein degradation. For this experiment, we used rat H4G1.1c3 cells stably expressing the DRE-driven GFP protein [26]. Treating these cells with a prototypical AHR ligand such as BaP and FICZ for 12 h caused the GFP expression that could be captured by fluorescence microscopy (Figure 6). This GFP fluorescence was suppressed in the presence of an AHR antagonist CH223191, supporting that the fluorescence corresponded to the AHR ligand-activated GFP expression. We observed that 100 μM 6-AN, which caused AHR protein degradation at that concentration, did not show any more fluorescence than the DMSO vehicle control, supporting that 6-AN is not an AHR ligand. We also determined that both MG132 (10 μM) and GA (0.1 μM) did not cause GFP expression and are not AHR ligands; however, all of them (6-AN, MG132, and GA) can reduce AHR protein levels (Figure 2A,E and Figure 5A).

#### 2.1.8. AHR Promotes Migration of A549 Cells in a Wound Healing Assay

There have been conflicting reports on the role of AHR in cell migration and metastasis, particularly on lung epithelial cells, in which some researchers showed suppression of migration and metastasis by AHR using the knockdown approach in A549 cells [14]. Here, we used our *AHR* knockout (5G11) A549 cells to determine the role of AHR on cell migration in a wound healing assay. We observed that A549 cells migrated slower when the AHR protein was absent (Figure 7A). Promoting CMA by 100 μM 6-AN treatment of A549 cells for up to 48 h showed similar retardation of migration as observed in *AHR* knockout 5G11 cells. This inhibition of cell migration by 6-AN was expected since 6-AN was shown to suppress the migration of acute myelogenous leukemia cells by inhibiting enzymes involved in the pentose phosphate pathway [38]. However, this suppression of migration by 6-AN was abolished in *AHR* knockout 5G11 cells, supporting that this 6-AN effect on A549 cells is AHR-dependent.

#### 2.1.9. AHR Promotes EMT in A549 Cells

Next, we examined the role of AHR on the EMT of A549 cells. We measured the transcript and protein levels of an epithelial marker, E-cadherin, and the mesenchymal markers vimentin and N-cadherin in wild-type and *ahr* KO 5G11 A549 cells. We observed that the transcript levels of *E-cadherin* and *Vimentin* were 2.6- and 0.5-fold, respectively, in *AHR* knockout 5G11 cells when compared to the wild-type A549 cells (Figure 7B), although the reduction in the *vimentin* transcript was not significant. The Western blot results showed that E-cadherin and vimentin proteins had the same trend as the transcripts, with a clear increase in E-cadherin and a decrease in vimentin in a statistically significant manner (Figure 7C). These results clearly supported the fact that AHR favors the mesenchymal phenotype in A549 cells. However, there was no change in the *N-Cadherin* transcript and protein levels in the presence or absence of AHR (Figure 7B,C). Treatment of A549 cells with 100 μM 6-AN for 24 h did not seem to alter the E-cadherin and vimentin protein levels in A549 cells, and lacking AHR in 5G11 did not reveal any trend of the 6-AN effect that was AHR-dependent (Figure 7C). Realizing that there should be about 50% of AHR content after this 6-AN treatment, it is conceivable that 50% of AHR content might be sufficient to maintain the levels of these markers, and 6-AN might very well elicit some AHR-independent effect that complicated the picture. Next, we were interested to see any effect on these markers when the AHR content was increased < 2-fold by knocking down LAMP2 in A549 cells (Figure 3A). We observed that vimentin was upregulated in LAMP2 knockdown A549 cells when compared to the wild-type A549 cells (Figure 7D), consistent with the predicted pattern when AHR was upregulated. However, E-cadherin was also upregulated, showing a more complex picture that might be complicated by the inhibition of the CMA-mediated degradation of an EMT inhibitor when LAMP2 was downregulated. Downregulation of LAMP2 might also trigger partial EMT, causing cells to co-express both epithelial and mesenchymal markers. During partial EMT, the retained epithelial markers (such as E-cadherin) were shown to cluster cancer cells before migration [39]. Interestingly, upregulation of E-cadherin expression might not be sufficient to block invasion, such as in the case of pancreatic cancer [40]. When we repeated the LAMP2 knockdown experiment using *AHR* knockout 5G11 cells, we observed that E-cadherin and vimentin were essentially unchanged when LAMP2 was knocked down, whereas a slight increase in N-cadherin levels was observed (Figure 7E). When comparing these results with the *AHR* knockout results in Figure 7C, knocking down LAMP2 appeared to suppress E-cadherin and increase vimentin levels, revealing a mesenchymal phenotype that was LAMP2-dependent but not AHR-dependent.

To directly address the migration and invasion potential mediated by AHR, we conducted Transwell assays to determine the migration and invasion potential of the wild-type and *AHR* knockout A549 cells. We observed that AHR significantly promoted cell migration and invasion by 2.1- and 1.8-fold, respectively, when we compared the *AHR* knockout 5G11 with the wild-type A549 cells (Figure 7F,G). 6-AN similarly suppressed the migration and invasion of A549 cells by 1.9- and 3.2-fold, respectively, in an AHR-dependent manner since this 6-AN effect was abrogated in the *AHR* knockout cells. Collectively, AHR clearly drives the invasion and migration of A549 cells. Activating the CMA degradation of AHR by 6-AN effectively slows down the invasion and migration of A549 cells in a mechanism that cannot be fully explained by E-cadherin and vimentin levels.

## 3. Discussion

Without the addition of exogenous ligand, AHR is subjected to autophagy in several human cell lines. Two autophagic mechanisms, namely selective macroautophagy and CMA, can selectively degrade client proteins such as human AHR. Here, we provide evidence that AHR undergoes CMA regularly in A549 cells without the addition of any exogenous ligand (Figure 8). This CMA-mediated AHR degradation can be activated by 6-AN in a dose- and time-dependent manner. Down-regulation of LAMP2A or mutation of the CMA motif of AHR abrogates this effect, strongly supporting the role of CMA in degrading AHR in A549 cells. Importantly, altering the degradation of AHR clearly affects its activities in the ligand-activated gene transcription and modulation of the EMT of A549 cells. In the case of HeLa cells, downregulation of LC3 impairs the degradation of AHR, supporting that selective macroautophagy is responsible for AHR protein degradation [20]. Interestingly, unlike A549 cells, treatment with 100 μM 6-AN for 24 h does not change the AHR protein levels in HeLa cells [20], revealing that selective macroautophagy, but not CMA, degrades AHR in a different human cell type. However, we also observed that AHR undergoes LAMP2A-mediated degradation in the triple-negative MDA-MB-468 human breast cancer cells [26], revealing that AHR can undergo different autophagy mechanisms in a cell-specific manner. There is precedent in the literature that maintaining cellular protein levels via autophagy can be important. For example, CMA is responsible for the degradation of SMAD3 [41], Erk3 [42], Dicer [43], and oxidized PRL2 [44]. Analogous to human AHR, Tau and α-synuclein have been reported to undergo both selective macroautophagy and CMA [45,46,47]. Although AHR can be degraded via either CMA or selective macroautophagy in a cell-line-specific manner, it is yet unclear how different human cells select which of the two autophagy mechanisms, or both, to degrade AHR.

Targeting autophagy-dependent AHR degradation can potentially be an effective therapeutic approach. For example, activation of AHR in the gut can be beneficial for the treatment of inflammatory bowel disease [48]. Thus, AHR agonists might be effective for this treatment. Interestingly, P140, a 21mer phosphopeptide derived from U1-70K spliceosomal protein, suppresses CMA and is proposed to be a mechanism for the treatment of inflammatory bowel disease [49]. Suppression of CMA might increase the AHR protein levels in the gut and thereby elicit a synergistic effect of AHR activation when used in combination with an AHR agonist—an interesting regimen for the treatment of inflammatory bowel disease.

The carboxy terminus of Hsc70-interacting protein (CHIP), a co-chaperone of HSC70, contains a U-box ring-finger motif found in ubiquitin ligases. This CHIP has been implicated in promoting proteasomal degradation of CFTR [50], tau [51], and hypoxia-inducible factor 1 alpha (HIF-1α) [52]. It is well known that binding HSC70 to the CMA motif of proteins escorts proteins to LAMP2A at the lysosomal membrane, followed by the internalization of proteins for degradation via CMA. Interestingly, HIF-1α also undergoes CMA, and CHIP is required for the interaction of HSC70 and HIF-1α interaction [53]. Like HIF-1α, AHR has been shown to interact with CHIP in vitro [54]. It is conceivable that CHIP may be involved in the degradation of AHR via CMA.

Although it has been widely accepted that AHR ligands and GA cause degradation of AHR via the ubiquitin–proteasome pathway, it appears that autophagy might also be involved in determining the AHR levels after ligand or GA treatment in A549 cells (Figure 8). Clearly, proteasomal degradation is primarily responsible for the degradation of AHR within 2 h of ligand or low-dose GA treatment in A549 cells. However, in time, autophagy becomes active in degrading AHR in A549 cells, suggesting that degradation of AHR via autophagy may be temporarily interrupted soon after a low dose of GA treatment, which activates proteasomal degradation of AHR. A high dose of GA (1 μM), however, promotes AHR degradation via autophagy in A549 cells.

Glutathione peroxidase 4 (GPX4) is known to be a CMA substrate. Inhibition of CMA increases GPX4 protein levels, leading to the inhibition of ferroptosis in mouse hippocampal HT-22 cells [55]. It has been reported that increased AHR protein levels inhibit ferroptosis in human lung adenocarcinoma PC-9 cells by increasing the expression of SLC7A1, a molecule that suppresses ferroptosis by reducing the reactive oxygen species content [56]. Interestingly, AHR is also a CMA substrate, and like GPX4, inhibition of the CMA-mediated degradation of AHR can be a viable approach for the regulation of ferroptosis.

Resveratrol has been reported to inhibit ligand-activated AHR transcriptional activity in a manner that does not act as a typical AHR antagonist since no binding of resveratrol to AHR was observed [57]. Nevertheless, resveratrol inhibits AHR function. Interestingly, resveratrol also inhibits the transforming growth factor beta 1-mediated EMT by downregulating vimentin and upregulating E-cadherin in A549 cells [58]. This is the same outcome as observed in the *AHR* knockout A549 cells, suggesting that inhibition of the AHR function by resveratrol may be, in part, responsible for EMT inhibition in A549 cells.

Although there are conflicting data in the literature showing the effect of AHR on the EMT in A549 cells, we clearly observed that AHR promotes the EMT in these cells via the CRISPR/Cas9 knockout approach. When the *ahr* gene is disrupted in A549 cells with no AHR protein production, E-cadherin is clearly upregulated, whereas vimentin is also clearly downregulated, supporting the epithelial phenotype when AHR is absent. Wound healing, invasion, and migration experiments all unambiguously support the fact that AHR drives the EMT in A549 cells. Our finding is consistent with other researchers reporting that knocking down AHR in A549 cells suppresses invasion and migration potential [56]. Additionally, AHR drives non-small cell lung cancer tumorigenesis when the AHR protein is stabilized after deubiquitination by ubiquitin carboxy-terminal hydrolase isozyme L3 [59], and this AHR action of cancer tumorigenesis may involve Jak2/STAT3 signaling [60].

When we attempted to modulate AHR protein levels by either 6-AN or LAMP2 knockdown, we were unable to see any consistent trend of E-cadherin, vimentin, and N-cadherin expression that was AHR-dependent. Naturally, we must keep in perspective the non-AHR dependent effects of 6-AN and LAMP2 knockdown on EMT marker expression. In the case of LAMP2 knockdown, mechanisms such as inhibition of the transcriptional activation of an EMT inducer that favors EMT and upregulation of some EMT inhibitor that favors the epithelial phenotype could be involved. Examining more than three markers of tumor invasion may be necessary in this case. For example, the levels of matrix metalloproteinase (MMPs), actin cytoskeleton proteins, and cell–extracellular matrix interaction molecules can be measured to possibly provide a better picture of how AHR drives EMT [61,62].

Exploring how modulation of the AHR protein levels may alter cancer stem cell-like properties and associated gene expression can be insightful since overexpression of AHR has been implicated in an aggressive tumor phenotype in non-small cell lung cancer [60]. Nevertheless, treatment with 6-AN clearly suppresses the invasion and migration of A549 cells in an AHR-dependent manner. Although looking at the EMT marker levels alone fails to explain how a reduction in AHR levels (by 6-AN) favors the epithelial phenotype of A549 cells, modulation of AHR protein levels can be a viable approach in controlling AHR’s function.

## 4. Materials and Methods

### 4.1. Reagents and Antibodies

CQ, BaP, GA, 6-AN, puromycin, CH223191, crystal violet, PMSF, and leupeptin were purchased from Sigma-Aldrich (St. Louis, MO, USA). (S)-MG132 and FICZ were purchased from Cayman Chemical (Ann Arbor, MI, USA). pLKO.1 Lentiviral LAMP2 shRNA plasmids and pLKO.1 Lentiviral ATG5 shRNA plasmids were purchased from Dharmacon (Lafayette, CO, USA). pGFP^2^-N2-AHR(NEKFF) and pGFP^2^-N2-mutant AHR(NAKAF) plasmids were previously generated by our lab [26]. pCMV-VSV-G was a gift from Bob Weinberg (Addgene plasmid # 8454; RRID: Addgene_8454). pCMV-dR8.2 dvpr was a gift from Bob Weinberg (Addgene plasmid #8455; RRID: Addgene_8455). EndoFectin transfection reagent was purchased from GeneGopoeia (Rockville, MD, USA). TRI Reagent and a Direct-zol RNA miniprep kit were purchased from Zymo Research (Irvine, CA, USA). MMLV high-performance reverse transcriptase was purchased from Epicentre (Madison, WI, USA). iTaq SYBR green supermix was purchased from Bio-Rad (Hercules, CA, USA). Dynabeads Protein G was purchased from Invitrogen (Carlsbad, CA, USA). EnGen Spy Cas9 NLS was purchased from NEB (Ipswich, MA, USA), multi-guide sgRNA was purchased from Synthego (Redwood, CA, USA), Lipofectamine RNAiMAX was purchased from Thermo Fisher Scientific (Waltham, MA, USA), and QuickExtract DNA extraction solution was purchased from Lucigen (Middleton, WI, USA). PCR Master Mix was purchased from Promega (Madison, WI, USA), and Falcon 24-well cell culture inserts with transparent PET membranes (8.0 μm pore size) and Matrigel matrix were purchased from Corning Inc. (Corning, NY, USA). FBS (HyClone), DMEM (HyClone), and Opti-MEM reduced serum medium were purchased from Thermo Fisher Scientific (Waltham, MA, USA). GlutaMAX-I and penicillin–streptomycin were purchased from Invitrogen (Carlsbad, CA, USA). Rabbit anti-AHR polyclonal antibody (SA210) was purchased from Enzo Life Sciences (Farmingdale, NY, USA). Mouse anti-LAMP2 monoclonal antibody (H4B4), mouse anti-Hsc70 monoclonal antibody (B-6), mouse anti-E-cadherin monoclonal antibody (G-10), mouse anti-Vimentin monoclonal antibody(V9), and mouse anti-N-cadherin monoclonal antibody (13A9) were purchased from Santa Cruz Biotechnology (Dallas, TX, USA). Rabbit anti-ATG5 polyclonal antibody (2630) was purchased from Cell Signaling Technology (Danvers, MA, USA). Donkey anti-rabbit and donkey anti-mouse secondary antibody conjugated with IRDye 680 or 800CW, Revert 700 Total Protein Stain, and nitrocellulose membrane were purchased from LI-COR Bioscience (Lincoln, NE, USA).

### 4.2. Cell Culture

A549 cells were a gift from Dr. John Livesey (University of the Pacific) and were authenticated by ATCC before being used for experiments in this paper. AD293 cells were purchased from Agilent Technologies (Santa Clara, CA, USA). Rat H4G1.1c3 stable cells carrying a DRE-driven GFP cDNA were a gift from Dr. Michael Denison (University of California, Davis). All the cell lines were cultured in DMEM supplemented with 10% fetal bovine serum (FBS), 1% penicillin–streptomycin, and 1% GlutaMAX-I at 37 °C, 5% CO_2_.

### 4.3. Preparation of Whole Cell Extract and Western Blot Analysis

A549 cells were scraped in cold PBS and centrifuged at 400× *g* for 5 min to collect cell pellets. The collected cells were washed once with cold PBS and then lysed in lysis buffer (25 mM HEPES pH7.4, 0.4 M KCl, 1 mM EDTA, 1 mM DTT, 10% glycerol, 1% NP40, 1 mM PMSF, and 2 μg/mL leupeptin). After 3 cycles of freezing and thawing, cell lysates were kept on ice for 30 min and then centrifuged at 16,000× *g* for 15 min at 4 °C. The supernatants were used as whole-cell extracts. Total protein concentrations were measured by BCA assay. Proteins in 20 μg of whole cell extract from each sample were separated by 12% SDS-PAGE and then transferred to nitrocellulose membranes via the wet transfer method. Total protein staining was determined using LI-COR Revert 700 Total Protein Stain for normalization. Non-specific binding was blocked in a blocking buffer (PBS, 0.1% Tween-20, and 5% BSA) for 1 h. The primary antibodies and their dilution are as follows: 1:2000 for anti-AHR (SA210); 1:1000 for anti-ATG5 (2630); 1:200 for anti-LAMP2 (H4B4); 1:200 for anti-Hsc70 (B-6); 1:200 for anti-E-cadherin (G-10); 1:200 for anti-Vimentin (V9); and 1:200 for anti-N-cadherin (13A9). After washing with PBST (PBS and 0.1% Tween-20) 5 times, the nitrocellulose membrane was incubated in 1:10,000 dilution of donkey secondary antibody conjugated with IRDye 680 or 800 CW. Results were obtained and quantified using a LI-COR Odyssey CLx imaging system (Lincoln, NE, USA).

### 4.4. RNA Extraction and Reverse Transcription-Quantitative Polymerase Chain Reaction (RT-qPCR)

Total RNA was extracted from A549 cells using TRI Reagent (Zymo Research) and an RNA miniprep kit (Direct-zol) according to the manufacturer’s recommendations. cDNA was reverse transcribed from 1 μg of RNA using MMLV high-performance reverse transcriptase (Epicentre) into a final volume of 20 μL cDNA solution, and 1 μL of it was used as the qPCR template. qPCR was performed with iTaq SYBR green supermix (Bio-rad, USA) on a CFX Connect real-time PCR operating system (Bio-rad, USA) according to the following protocol: 95 °C for 2 min, 40 cycles of 95 °C for 15 s, and 60 °C for 1 min. Relative gene expression was analyzed by the 2^−ΔΔCq^ method [63], and *18s* rRNA was used as an internal control for normalization. The primer sequences were as follows: *18s* forward: 5′-CGCCCCCTCGATGCTCTTAG-3′ and reverse: 5′-CGGCGGGTCATGGGAATAAC-3′; *CYP1A1* forward: 5′-GGCCACATCCGGGACATCACAGA-3′ and reverse: 5′-TGGGGATGGTGAAGGGGACGAA-3′; *E-Cadherin* forward: 5′-GCCTCCTGAAAAGAGAGTGGAAG-3′ and reverse: 5′-TGGCAGTGTCTCTCCAAATCCG-3′; *Vimentin* forward: 5′-TGTCCAAATCGATGTGGATGTTTC-3′ and reverse: 5′-TTGTACCATTCTTCTGCCTCCTG-3′; *N-Cadherin* forward: 5′-ACAGTGGCCACCTACAAAGG-3′ and reverse: 5′-CCGAGATGGGGTTGATAATG-3′.

### 4.5. Generation of ATG5, LAMP2 Stable Knockdown A549 Cells Using Lentivirus

Lentivirus containing ATG5 or LAMP2 shRNA was prepared as follows: AD293 cells (7 × 10^5^) in 5 mL of growth media (10% fetal bovine serum and 2 mM GlutaMAX-I in DMEM) were seeded in a 25 cm^2^ flask. Cells were incubated at 37 °C and 5% CO_2_ overnight. Then, AD293 cells were transfected using 10 μL EndoFectin transfection reagent with 5 μg plasmids (2.5 μg of pLKO.1 specific shRNA plasmid, 1.875 μg of the pCMV-dR8.2 dvpr packaging plasmid, and 0.625 μg of the pCMV-VSV-G envelope plasmid). Fresh complete medium was replaced 15 h after transfection. After 24 h, the medium containing lentiviral particles was transferred to a 15 mL tube and stored at 4 °C. Another 5 ml of fresh complete medium was added to the cells, and the medium containing lentiviral particles was harvested after 24 h of incubation. The combined medium was centrifuged at 400× *g* for 5 min to pellet any AD293 cells, and the supernatant was used for infection. Stable ATG5 or LAMP2 knockdown cell lines were generated as follows: A549 cells were seeded in a 25 cm^2^ flask to achieve 50–70% confluent the next day. Fresh complete medium containing 8 μg/mL polybrene was replaced. A total of 500 μL of medium containing lentiviral particles was added into the flask. After 24 h, the medium was replaced with fresh complete medium containing 1.5 μg/mL of puromycin for stable cell line selection. This was replaced with fresh medium containing puromycin every 2–3 days. ATG5 knockdown stable A549 cells were generated using pLKO.1 Lentiviral (TRC) ATG5 shRNA #5 (TRCN0000151963) plasmid; LAMP2 knockdown stable A549 cells were generated using pLKO.1 Lentiviral (TRC) LAMP2 shRNA #4 (TRCN0262) plasmid.

### 4.6. CRISPR/Cas9-Mediated AHR Knockout in A549 Cells

Three different single guide RNAs (sgRNA) targeting exon 2 of the human *AHR* gene were used to knock out the *AHR* gene in A549 cells. The sequences were as follows: sgRNA1: 5′- GCTGAAGGAATCAAGTCAAA-3′; sgRNA2: 5′- ACAAGATGTTAT-TAATAAGT-3′; and sgRNA3:5′- GAGAGCCAAGAGCTTCTTTG-3′. Cas9 nuclease NLS and sgRNAs were introduced as ribonucleoprotein (RNP) complex into A549 cells through transfection using Lipofectamine RNAiMAX according to the manufacturer’s recommendations. In brief, 3 μM Cas9 nuclease NLS was combined with 3 μM sgRNAs (1 μM each of 3 sgRNAs) to form RNPs in 12.5 μL volume with the Opti-MEM. We gently mixed the reaction and incubated it at room temperature for 10 min. A total of 1.2 μL of transfection reagent RNAiMAX was diluted in 12.5 μL of the Opti-MEM and was added directly into the RNP tube. The RNPs/liposome complexes were mixed gently and incubated at room temperature for 20 min. Meanwhile, a 3.2 × 10^5^ cells/mL A549 cell suspension was prepared, and 125 μL of it was added into each well of a 96-well plate, followed by mixing it with 25 μL of RNPs/liposome complexes. The transfected cells were incubated at 37 °C and 5% CO_2_ for 72 h. Then, the isolation of single cells from the knockout cell pool was accomplished through limiting dilution according to the protocol from Synthego. A total of 0.5–1 cell/100 μL of the diluted cell suspension was seeded into each well of a 96-well plate. To genotype clones, genomic DNA was isolated using QuickExtract DNA extraction solution (Lucigen). PCR was performed to amplify the edited region using PCR Master Mix (Promega) with the following primers: OL921 forward 5′-TCGGAAGAATTTAACC-CATTCCCT-3′ and OL922 reverse 5′-TGCAGCCACTGAAATGATGC-3′. A DNA fragment ~500 bp was observed by agarose gel electrophoresis and was purified for Sanger sequencing (Functional Biosciences, WI, USA). The sequencing data were uploaded to the online Inference of CRISPR Edits (ICE) v3.0 analysis tool (http://ice.synthego.com, accessed on 8 September 2023) for knockout analysis.

### 4.7. Transient Transfection

A549 cells were seeded in a 6-well plate at 90–95% confluency at the time of transfection. Plasmid DNA, EndoFectin transfection reagent, and Opti-MEM were equilibrated to room temperature before use. Cells were transfected with 4 μg of plasmid and 8 μL of transfection reagent. Both plasmids and transfection reagents were diluted in the Opti-MEM. Then, the diluted transfection reagent and the diluted DNA were combined and kept at room temperature for 20 min to allow DNA–transfection reagent complexes to form. The combined complexes were added to each well and mixed gently. The cells were harvested for analysis after 48 h of incubation at 37 °C and 5% CO_2_.

### 4.8. Co-Immunoprecipitation

A549 cells were cultured in a 75 cm^2^ flask and reached 90–95% confluency at the time of the experiment. Cells were treated with or without CQ for 6 h and then were harvested to be lysed in lysis buffer (25 mM HEPES pH7.4, 0.15 M KCl, 1 mM EDTA, 1 mM DTT, 10% glycerol, 10 mM N-Ethylmaleimide, 1 mM PMSF, and 2 μg/mL leupeptin). About 2 mg of the whole cell extract was incubated with rabbit anti-AHR antibody (SA210) for 30 min at room temperature. Then, the samples were added to the pre-equilibrated Dynabeads Protein G (Invitrogen) and q.s. to 1 mL with IP buffer (25 mM HEPES pH7.4, 1 mM EDTA, 1 mM DTT, 10% glycerol, 150 mM NaCl, 0.05% Tween-20, and 1 mg/mL BSA). The samples were rotated at 60 rpm in a cold room for 16–18 h. The magnetic beads-Ab-Ag complex was washed 3 times with cold IP buffer on the magnet. The magnetic beads-Ab-Ag complex was resuspended in 30 μL of electrophoresis sample buffer and boiled at 95 °C for 3 min to free the bound protein. A total of 1% of the whole cell extract was used as an input control. All the samples were analyzed by Western blot with antibodies against AHR, LAMP2, and Hsc70.

### 4.9. Ligand Dependent, DRE-Driven Expression of GFP in H4G1.1c3 Cells

H4G1.1c3 cells (3 × 10^5^) were seeded into each well of a 24-well plate. After incubation at 37 °C and 5% CO_2_ for 24 h, the cells reached 70–85% confluence. A total of 1 mL of fresh complete medium was exchanged for each well. The cells were then treated with DMSO (0.2%), 6-AN (100 μM), MG132 (10 μM), GA (100 nM), BaP (5 μM), FICZ (1 μM), CH223191 (10 μM), BaP plus CH223191, and FICZ plus CH223191 and incubated at 37 °C and 5% CO_2_ for 12 h. Fresh complete medium was exchanged after treatment. Fluorescence images were acquired by a Keyence BZ-X700 fluorescence microscope in 4× objective.

### 4.10. Wound Healing Assay

A549 cells were seeded in 6-well plates and formed monolayers at the time of wounding. A sterile 1 mL pipette tip was used to scratch across the monolayers to form a linear wound. Then, the disassociated cells and debris were removed by washing with PBS. Cells were treated with DMSO or 6-AN for 48 h. Representative images were taken at 0 and 48 h after the treatments at the same position under an inverted microscope (BZ-X700, KEYENCE, Itasca, IL, USA) with a camera. The scale bar on the representative images is 500 μm.

### 4.11. Transwell Migration and Invasion Assay

Falcon 24-well cell culture inserts with transparent PET membrane (8.0 μm pore size) and Corning Matrigel matrix (1:5 dilution) were used to determine the cell migration and invasion capability. For migration assay, 5 × 10^4^ cells were seeded into the upper chamber, and 700 μL of complete DMEM medium was added in the lower chamber and placed in 24-well plates. For the invasion assay, 1 × 10^5^ cells were seeded into the upper chamber, which was coated with Matrigel before use. After 24 h incubation, the cells on the inserts were fixed with methanol for 10 min and stained with 0.1% crystal violet for 5 min. Then, the cells on the top side of the membrane were removed with cotton swabs carefully. Only the cells that migrated or invaded through the membrane to the bottom of inserts were imaged using an inverted microscope (BZ-X700, KEYENCE, Itasca, IL, USA) with a camera. Three fields, which covered about 80% of the well, were randomly captured and were analyzed by ImageJ v1.53k software.

### 4.12. Statistical Analysis

GraphPad Prism 9 software (La Jolla, CA, USA) was used for statistical analysis. The statistical significance of the differences between group means was evaluated by one-way or two-way ANOVA using Tukey’s or Sidak’s test for multiple comparisons. Statistical significance is indicated as follows: * *p* < 0.05, ** *p* < 0.01, *** *p* < 0.001, **** *p* < 0.0001, and *p* > 0.05 (ns, not significant). The two-tailed unpaired *t*-test was used to determine the statistical significance in Figure 7D,E.

## 5. Conclusions

AHR undergoes CMA continuously in A549 cells. Activation of CMA by 6-AN degrades AHR, causing downregulation of AHR functions, such as the ligand-dependent activation of gene transcription and promotion of the EMT in A549 cells.

## Figures and Tables

**Figure 1 ijms-24-15116-f001:**
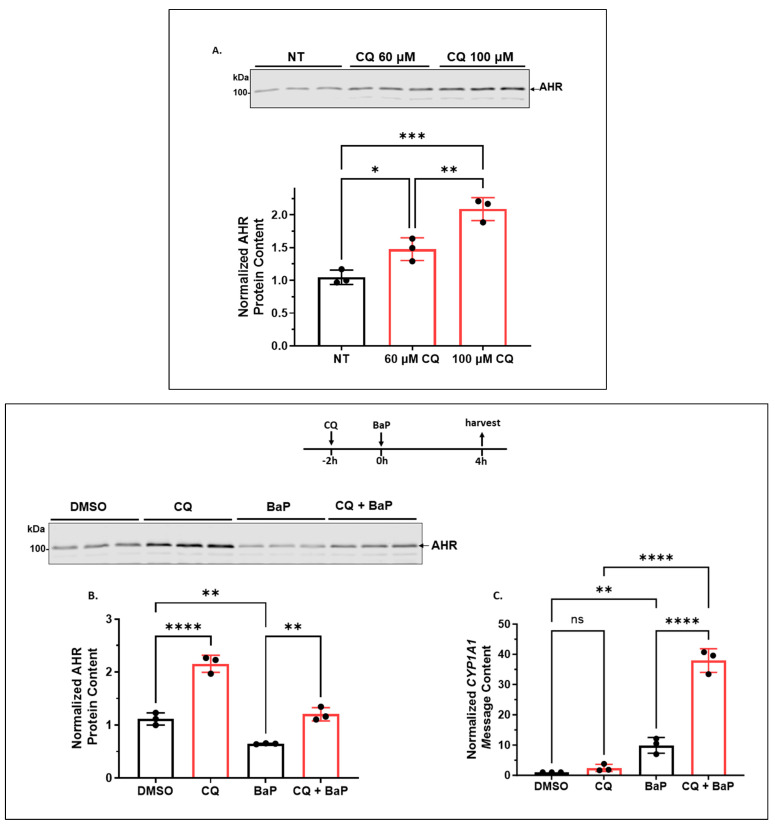
Inhibition of lysosomal degradation increased the AHR protein level and its ligand-dependent activation of the *CYP1A1* gene transcription in A549 cells. (**A**) Western blot results of cells treated with 0, 60 or 100 μM CQ for 6 h. The images above are biological triplicates of one experiment. (**B**) Western blot results of cells treated with 100 μM CQ for 6 h. At 2 h post-CQ treatment, cells were co-treated with DMSO or 5 μM BaP for the remaining 4 h. The images above are biological triplicates of one experiment. (**C**) RT-qPCR results of *CYP1A1* message level. Each experiment was in biological triplicate and was repeated once with similar results. The plots showed as the means with error bars (means ± SD, *n =* 3). The statistical significance of the differences between group means was evaluated by one-way ANOVA using Tukey’s test for multiple comparisons. Statistical significance is indicated as follows: * *p* < 0.05, ** *p* < 0.01, *** *p* < 0.001, **** *p* < 0.0001, and *p* > 0.05 (ns, not significant).

**Figure 2 ijms-24-15116-f002:**
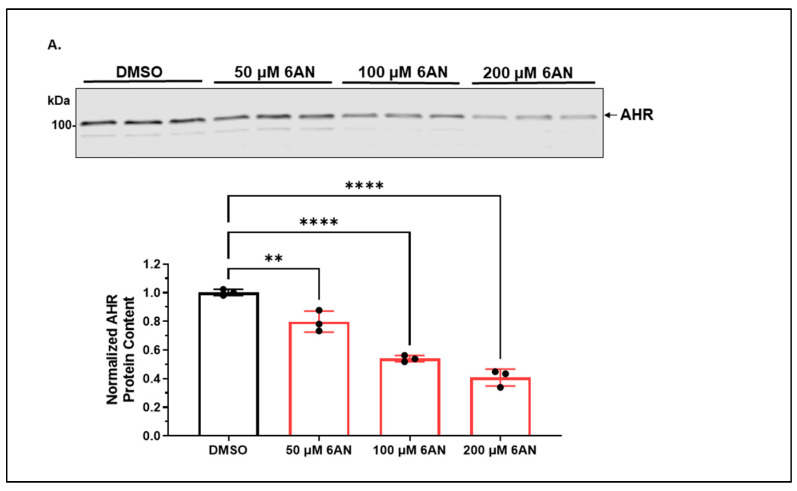
6-AN decreased the AHR protein level via CMA and suppressed the ligand-induced activation of its target gene transcription in A549 cells. (**A**) Western blot results of cells treated with DMSO or 6-AN (50, 100 or 200 μM) for 24 h. The images above are biological triplicates of one experiment. (**B**) Western blot results of cells treated with 100 μM 6-AN (**B**, **left panel**) or 5 μM BaP (**B**, **right panel**) for 0, 2, 6, 10, or 24 h. Each time point represents means ± SD, *n =* 3. (**C**) Western blot results of cells pretreated with DMSO or 100 μM 6-AN for 18 h, followed by 100 μM CQ treatment for an additional 6 h. The images above the plot are biological duplicates of one experiment. This experiment was repeated one more time to generate the plot. (**D**) Western blot results of cells pretreated with DMSO or 100 μM 6-AN for 18 h followed by 10 μM MG132 treatment for an additional 6 h. The images above the plot are biological triplicates of one experiment. This experiment was repeated once with similar results. (**E**) Western blot results of cells treated with DMSO, 10 μM MG132, 100 μM CQ, or 10 μM MG132 plus 100 μM CQ for 6 h. The images are representatives of the plotted data. All plots are the means with error bars (means ± SD, *n =* 3). (**F**) Western blot results of cells treated with 4 mM metformin (Met) for 0, 4 or 24 h. The plot represents the means with error bars (means ± SD, *n =* 5). The images above are biological triplicates of one experiment. The statistical significance of the differences between group means was evaluated by one-way ANOVA using Tukey’s test for multiple comparisons. (**G**) Western blot results of cells treated with 0.5 μM rapamycin (Rap) for 0, 4 or 24 h. The plot represents the means with error bars (means ± SD, *n =* 5). The images above are biological triplicates of one experiment. (**H**) RT-qPCR results of cells pretreated with DMSO or 100 μM 6-AN for 18 h, followed by BaP (5 μM) or FICZ (1 μM) treatment for another 6 h. The experiment was performed with biological triplicates and was repeated once with similar results. The plot represents the means with error bars (means ± SD, *n =* 3). The statistical significance of the differences between group means was evaluated by one-way ANOVA using Tukey’s test for multiple comparisons. Statistical significance is indicated as follows: * *p* < 0.05, ** *p* < 0.01, *** *p* < 0.001, **** *p* < 0.0001, and *p* > 0.05 (ns, not significant).

**Figure 3 ijms-24-15116-f003:**
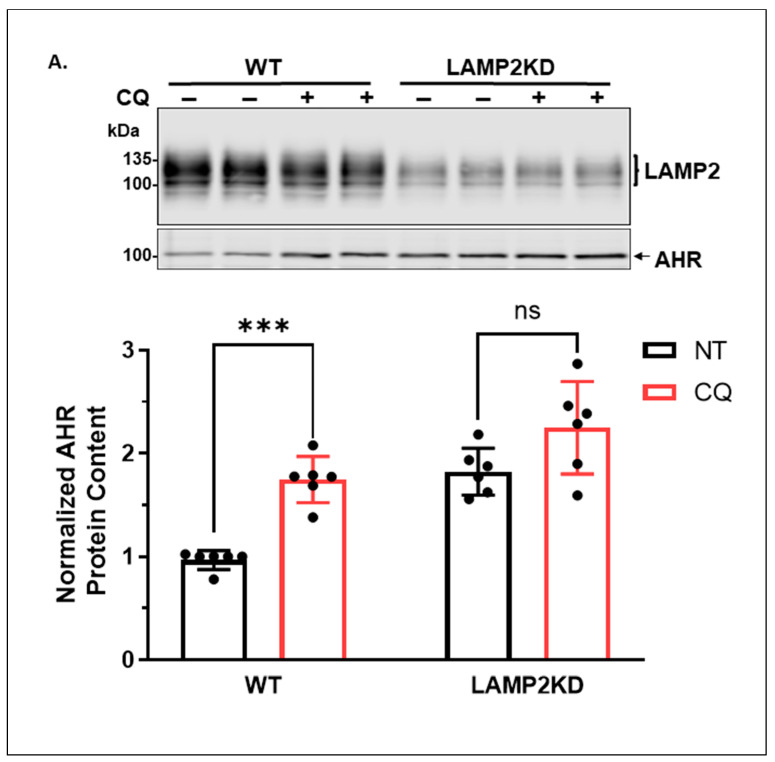
Lysosomal degradation of AHR required LAMP2A and the presence of its CMA motif in A549 cells. (**A**) Western blot results of A549 wild-type (WT) and LAMP2 stable knockdown (LAMP2KD) cells treated with or without 100 μM CQ for 6 h. Stable knockdown cells contain 30% of the wild-type LAMP2 content. Mature LAMP2, including LAMP2A, is highly glycosylated with a total molecular weight of about 100~130 kDa. The plot represents the means with error bars (means ± SD, *n =* 6). The images above are biological duplicates of one experiment. (**B**) Western blot results of A549 wild-type (WT) and ATG5 stable knockdown (ATG5KD) cells treated with or without 100 μM CQ for 6 h. Stable knockdown cells contain 28% of the wild-type ATG5 content. Intracellular ATG5 is conjugated with ATG12. In the Western blot, a band at ~55 kDa represents the ATG5-ATG12 complex. The images are biological triplicates from one experiment, and the experiment was repeated once with similar results. (**C**) CRISPR/Cas9-mediated knockout of *AHR* gene in A549 cells. Western blot analysis of AHR protein levels in wild-type (WT) and five CRISPR/Cas9 *AHR* knockout (KO) clones. Sequencing alignment showed that clones 4H2, 2F6, 3C9, and 5F2 are heterozygous compound knockouts. The plot indicates the INDEL efficiencies and knockout scores from the online Inference of CRISPR Edits (ICE) analysis. Clone 5G11 is a homozygous knockout with 47 nucleotides deleted in the exon 2 region of AHR genomic DNA, leading to a frameshift deletion. (**D**) Western blot results of cells transiently expressing GFP-AHR wild type (NEKFF) or GFP-AHR mutant (NAKAF) treated with DMSO or 6-AN for 24 h at 48 h post-transfection. The plot represents the means with error bars (means ± SD, *n =* 3). The experiment was performed with biological triplicates and was repeated once with similar results. The images represent the biological triplicates of one experiment. NP represents A549 cells that have undergone the transfection process without the GFP plasmid, whereas WT is the wild-type A549 cells. (**E**) Co-immunoprecipitation results of cells treated with or without 100 μM CQ for 6 h. Anti-AHR antibody SA210 was used for immunoprecipitation of AHR (IP). The 1% input represents 1% of the total protein lysate used for immunoprecipitation. The plot represents the means with error bars (means ± SD, *n =* 4). The images are representative of the plotted data. The statistical significance was evaluated by one-way (**E**) or two-way (**B**–**D**) ANOVA using Tukey’s test for multiple comparisons. Statistical significance is indicated as follows: * *p* < 0.05, ** *p* < 0.01, *** *p* < 0.001, **** *p* < 0.0001, and *p* > 0.05 (ns, not significant).

**Figure 4 ijms-24-15116-f004:**
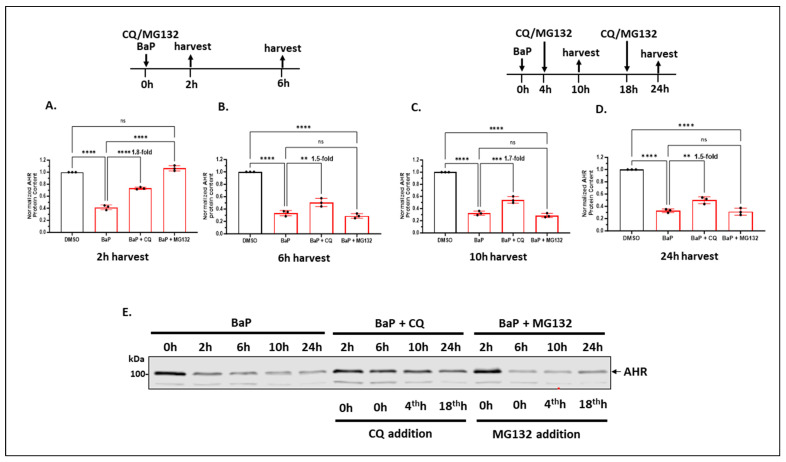
Lysosomal degradation of AHR was ongoing in the background while AHR underwent rapid proteasomal degradation after ligand treatment in A549 cells. Cells were treated with 5 μM BaP (or DMSO), 100 μM CQ (or water), and 10 μM MG132 (or DMSO) in different combinations, followed by Western blot analysis of the AHR content. Treatment conditions are diagrammed above the plots in (**A**–**D**). The plots represent the means with error bars (means ± SD, *n =* 3). The experiment was performed with biological triplicates and was repeated once with similar results. The statistical significance of the differences between group means was evaluated by one-way ANOVA using Tukey’s test for multiple comparisons. (**E**) A representative sample of the Western blot images of (**A**–**D**) in one Western blot analysis. Statistical significance is indicated as follows: ** *p* < 0.01, *** *p* < 0.001, **** *p* < 0.0001, and *p* > 0.05 (ns, not significant).

**Figure 5 ijms-24-15116-f005:**
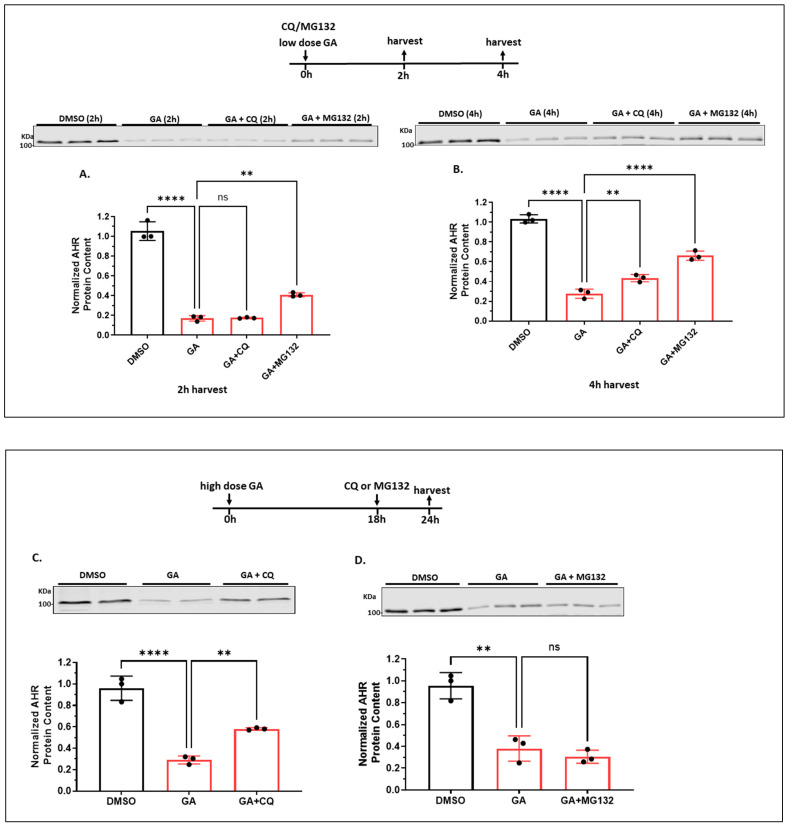
Lysosomal degradation of AHR occurred after A549 cells were treated with low or high doses of GA. Western blot results of cells treated with DMSO, 0.1 μM GA (low dose), 0.1 μM GA plus 100 μM CQ, or 0.1 μM GA plus 10 μM MG132 for 2 h (**A**) or 4 h (**B**). The images represent the biological triplicates of one experiment. (**C**) Western blot results of cells pretreated with 1 μM GA (high dose) for 18 h, followed by 100 μM CQ treatment for another 6 h. The images represent the biological duplicates of one experiment. (**D**) Western blot results of cells pretreated with 1 μM GA (high dose) for 18 h, followed by 10 μM MG132 treatment for another 6 h. The images represent the biological triplicates of one experiment. For (**A**–**D**), all plots represent the means with error bars (means ± SD, *n =* 3); all experiments were performed with biological triplicates and were repeated once with similar results; the statistical significance of the differences between group means are evaluated by one-way ANOVA using Tukey test for multiple comparisons. (**E**) A plot showing the Western blot results of cells treated with either a low dose of GA (0.1 μM) or a high dose of GA (1 μM) for 0, 2, 6, 10, or 24 h. Each time point represents means ± SD, *n =* 3. Statistical significance is indicated as follows: ** *p* < 0.01, **** *p* < 0.0001, and *p* > 0.05 (ns, not significant).

**Figure 6 ijms-24-15116-f006:**
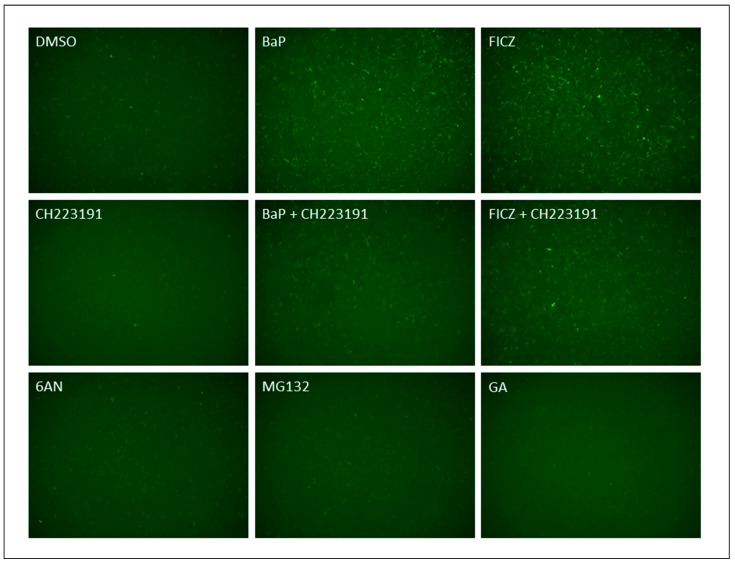
6-AN, (S)-MG132, and GA are not AHR ligands. Treatment with 6-AN, MG132, GA, BaP, FICZ, and CH223191 in rat H4G1.1c3 stable cells carrying a DRE-driven GFP cDNA. Cells were treated for 12 h with 0.2% DMSO, 100 μM 6-AN, 10 μM MG132, 100 nM GA, 5 μM BaP, 1 μM FICZ, 10 μM CH223191, 5 μM BaP plus 10 μM CH223191, or 1 μM FICZ plus 10 μM CH223191. The experiment was repeated two more times with similar results. Magnification: 4×.

**Figure 7 ijms-24-15116-f007:**
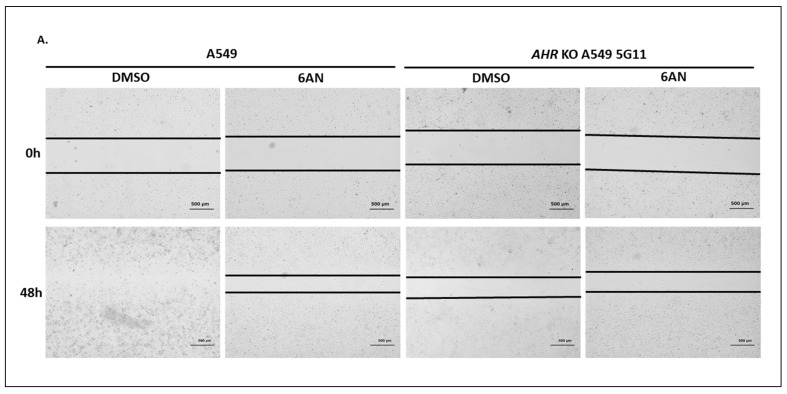
Suppression of the AHR protein level slowed down EMT in A549 cells. (**A**) Wound healing assay results showing the migration capacities of the wild-type and *AHR* knockout (KO, 5G11) A549 cells. Both cell lines were treated with DMSO or 100 μM 6-AN for 48 h. The microscopy images of wound closure were captured at 0 and 48 h after wounding. The black lines represent the area lacking cells. Scale bars, 500 μm. The experiment was repeated two more times with similar results. Magnification: 4×. (**B**) RT-qPCR results of the message levels of *E-cadherin, N-cadherin,* and *vimentin* in wild-type and *AHR* knockout (KO, 5G11) A549 cells. The plot represents the means with error bars (means ± SD, *n =* 3). The statistical significance of the differences between group means was evaluated by two-way ANOVA using Sidak’s test for multiple comparisons. Western blot results of E-cadherin, N-cadherin, and vimentin in (**C**) *AHR* knockout (KO, 5G11), (**D**) LAMP2 stable knockdown (LAMP2KD), and (**E**) LAMP2 stable knockdown (LAMP2 KD) of *AHR* knockout (KO) 5G11 A549 cells are shown as biological triplicates in one blot. Corresponding images in A549 wild-type cells are shown as the controls. 6-AN condition in C was performed with 100 μM 6-AN for 24 h. (**F**) Transwell migration assay results of the wild-type and *AHR* knockout (KO, 5G11) A549 cells treated with DMSO or 6-AN for 24 h. (**G**) Transwell invasion assay results of wild-type and *AHR* knockout (KO, 5G11) A549 cells treated with DMSO or 100 μM 6-AN for 24 h. Representative images of migrated and invaded cells were captured. Scale bars, 200 μm. The numbers of migrated and invaded cells were quantitated by ImageJ v1.53k software. All plots (**B**–**G**) represent the means with error bars (means ± SD, *n =* 3) and were repeated once with similar results. The statistical significance of the differences between group means was evaluated by two-way (**B**,**C**,**F**,**G**) ANOVA using Tukey’s test for multiple comparisons. The statistical significance of the differences between group means was evaluated by unpaired *t*-test (**D**,**E**). Statistical significance is indicated as follows: ** *p* < 0.01, *** *p* < 0.001, **** *p* < 0.0001, and *p* > 0.05 (ns, not significant).

**Figure 8 ijms-24-15116-f008:**
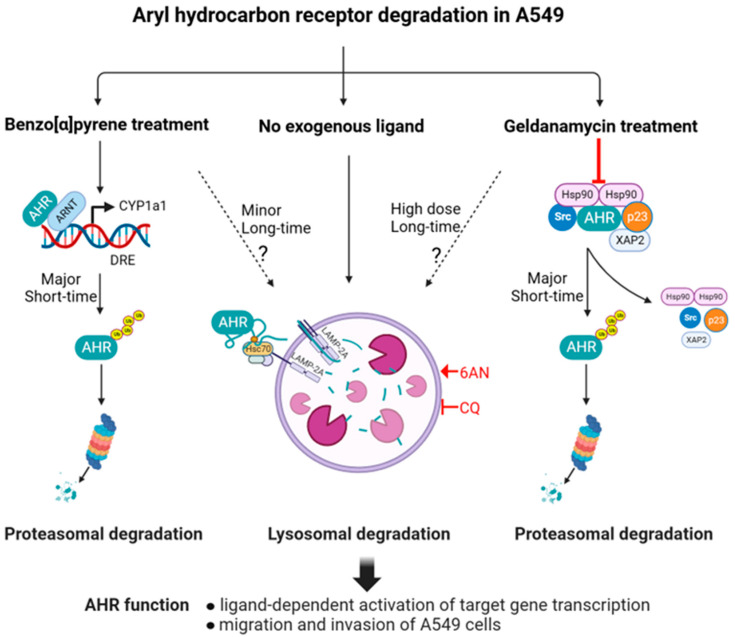
A diagram of CMA-mediated degradation of AHR in A549 cells. This figure was created with BioRender.com, www.biorender.com (accessed on 8 September 2023).

## Data Availability

Data are available upon request.

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
