# Peer review of "Activation of Chaperone-Mediated Autophagy Inhibits the Aryl Hydrocarbon Receptor Function by Degrading This Receptor in Human Lung Epithelial Carcinoma A549 Cells"

_ijms, 2023, doi:10.3390/ijms242015116_

Round 1

Reviewer 1 Report

The manuscript entitled:  Activation of chaperone-mediated autophagy inhibits the aryl  hydrocarbon receptor function by degrading this receptor in  human lung epithelial carcinoma A549 cells“ investigates connection between chaperone-mediated autophagy (CMA) and arylhydrocarbon receptor (AhR) level, potentially activity.

Despite interesting topic that investigates less known part of AhR biology, there are some uncertainties that should be specified, modified or improved.

 Namely:

 In general, in abstract, as well as throughout the manuscript, there are abbreviations that are not explained as they appear for the first time in the text – e.g. abstract (LAMP2, ATG5, MG132), FICZ (line 33 – no defined anywhere in the manuscript), CQ (line 82), NEKFF, HSC70 (lines 159, 160), GA (line 210) – it should be shortly defined what it is.

 Citations should be fixed as they contain a cross and a number. Neither both should be there.

Since most data in the publication deal with human AhR and CYP1A1, their genes should be written in upper case letters and italics (e.g. CYP1A1).

No loading controls (e.g. actin, tubulin, GAPDH) are present despite whole manuscript is based on western blotting technique. Moreover, the original images for Blots/Gels do not contain original blots/images but same cropped figures presented in the manuscript.

Subfigures of a given figure presented in the manuscript should be merged into one figure to obtain better lucidity.

Inappropriate mention about kynurenine – line 33 – despite often cited, it is not likely ligand, but pro-ligand of a super potent condensation product (S.-H. Seok, Z.-X. Ma, J.B. Feltenberger, H. Chen, H. Chen, C. Scarlett, Z. Lin, K. A. Satyshur, M. Cortopassi, C.R. Jefcoate, Y. Ge, W. Tang, C.A. Bradfield, Y. Xing, Trace derivatives of kynurenine potently activate the aryl hydrocarbon receptor (AHR), J. Biol. Chem. 293 (6) (2018) 19942005.) – therefore a better example of tryptophan-derived ligand of AhR should be used as an example next to FICZ.

There is inadequate interpretation of data for figure 1B, C – line 96 - ….showing that a 2-fold increase of the AHR content caused significant increase of the AHR function .“ – Figure 1C displays synergistic activity of CQ and BaP at CYP1A1 mRNA level. On the other hand, Figure 1B shows that combination of CQ and BaP does not affect AhR protein level at al. It is equal to negative control, DMSO. A 2-fold increase is observed for CQ only, but without significant impact on CYP1A1 expression. Thus, there is an impact on AhR activity (function) but not protein level.

Line 101 – number “6“ from 6AN is a part of a subheadings, 6AN is not introduced for the very first time

I suggest for most sentences containing a word “treatment” to use the preposition “with”, i.e. Treatment with MG132/6AN/CQ..” – instead of “Treatment of MG132/6AN/CQ,..”.  – lines 153.

Description of figures describing WB data often says: “The images represent the biological triplicates of one experiment.” – However, I believe, this is not correct. If one experiment equals to one passage of cells, then presented data are technical triplicate of one experiment. The authors just divided cellular lysate from one passage. This is technical replicate. I biological one would be from different passage of the cells.

Line 251 – since all experiments were performed in human A549 cell line and the authors detected CYP1A1  mRNA (Figure 1C), it is difficult to understand why there were used rat cells containing rat AhR which may have at least different affinity next to human AhR (which is well described phenomenon).

For better orientation, Figure 7C should be aligned similarly to transcripts (Fig 7B) – E-cadherin, vimentin, N-cadherin. Moreover, transcripts after 6AN treatment would be great to demonstrate as well. In Figure 7D – names of the proteins at the right should be closer to data, since e.g.Vimentin seems to belong to middle picture.

At line 244 it is stated: “Results from the time-dependent experiment showed that high dose of GA suppressed the AHR protein levels more effectively than low dose of GA (Fig. 5E), supporting that high dose of GA promotes AHR degradation via autophagy. However, we cannot rule out the possibility that higher dose of GA may also cause an increase of the proteasomal degradation of AHR.”  – Since there are no data that would demonstrate that higher dose of GA is a better autophagy inducer in A549 cell line, authors should either reformulate the sentences or detect e.g. LC3-I, II after GA treatment or ubiquitin on AhR as a proof of GA-induced proteasomal degradation (e.g. by Co-IP).

Interpretation at Line 240  ”We observed that CQ increased the AHR levels from 29 to 58% whereas MG132 decreased from 38% to 31% (Fig. 5C and D), showing that autophagy, but not proteasomal degradation, is essential for maintaining the AHR protein levels. “ should be corrected. A minimal difference without significance actually means, these values are equal. If authors want to keep such interpretation than at least 3 other measurementas should be performed to get significance.

Interpretation at line 242 “ The slight reduction of AHR levels by MG132 in this case is probably due to the activation of autophagy by MG132, as observed in Fig. 2D and E.” – is not correct. Figures 2D/2E do no show MG132-triggered autophagy, they demonstrate impact of MG132 on AhR content. Authors should demonstrate at least conversion of LC3-I to LC3-II for MG132 to be sure that MG132  induced autophagy in A549 cells.

The interpretation at lines 206-208 “The AHR protein levels were similar from 6h up to 24h after BaP treatment and similar inhibition of AHR autophagy by CQ was observed within that period (1.5 to 1.8-fold), suggesting that autophagy was involved for maintaining the AHR levels after ligand treatment (Fig. 4A-E).” is confusing. It is not clear to what sample(s) the folds are related to. Moreover, schematic figure should be present above WB representative image as the whole figure is difficult to understand.

The description of the data at line 194 is difficult to understand: “ Cotreatment of BaP with 100 M CQ, however, increased the AHR protein levels to 1.8-fold,… “ - compared to what? - This is not specified.

 Based on above comments, I do not recommend manuscript in the current form for publication until evident improvements has been made.

Author Response

Responses to Reviewer 1 comments (Reviewer’s comments is in italic):

The manuscript entitled, “Activation of chaperone-mediated autophagy inhibits the aryl hydrocarbon receptor function by degrading this receptor in human lung epithelial carcinoma A549 cells” investigates connection between chaperone-mediated autophagy (CMA) and aryl hydrocarbon receptor (AhR) level, potentially activity. Despite interesting topic that investigates less known part of AhR biology, there are some uncertainties that should be specified, modified, or improved.

  1. In general, in abstract, as well as throughout the manuscript, there are abbreviations that are not explained as they appear for the first time in the text – e.g. abstract (LAMP2, ATG5, MG132), FICZ (line 33 – no defined anywhere in the manuscript), CQ (line 82), NEKFF, HSC70 (lines 159, 160), GA (line 210) – it should be shortly defined what it is.

Response: All names have been added in red to their abbreviations the first time they appeared on the manuscript.

  1. Citations should be fixed as they contain a cross and a number. Neither both should be there.

Response: All references are now formatted in the manuscript.

  1. Since most data in the publication deal with human AhR and CYP1A1, their genes should be written in upper case letters and italics (e.g. CYP1A1).

Response: We have made the AHR target gene cyp1a1 in lowercase and italic. When we refer to protein, such as AHR, we have made them in uppercase.

  1. No loading controls (e.g. actin, tubulin, GAPDH) are present despite whole manuscript is based on western blotting technique.

Response: Correct. We intentionally did not use any reference protein such as beta-actin or GAPDH for normalization because it is more accurate to use total protein of the lane for normalization. This method of normalization is optimal with the use of LICOR imager to determine the intensity of the bands of interest using near-IR secondary antibody. Line 736 under 4.3 states that all western bands are normalized by total protein stain.

  1. The original images for Blots/Gels do not contain original blots/images but same cropped figures presented in the manuscript.

Response: We have now included all the original uncropped data in PDF.

  1. Subfigures of a given figure presented in the manuscript should be merged into one figure to obtain better lucidity.

Response: It is challenging to group the data into one figure without compromising clarity. After much thought, we believe that the current form is still the best.  

  1. Inappropriate mention about kynurenine – line 33 – despite often cited, it is not likely ligand, but pro-ligand of a super potent condensation product (S.-H. Seok, Z.-X. Ma, J.B. Feltenberger, H. Chen, H. Chen, C. Scarlett, Z. Lin, K. A. Satyshur, M. Cortopassi, C.R. Jefcoate, Y. Ge, W. Tang, C.A. Bradfield, Y. Xing, Trace derivatives of kynurenine potently activate the aryl hydrocarbon receptor (AHR), J. Biol. Chem. 293 (6) (2018) 1994–2005.) – therefore a better example of tryptophan-derived ligand of AhR should be used as an example next to FICZ.

Response: As suggested by the reviewer, kynurenine has been removed in line 36 (which was line 33). Although the cyclized structure of kynurenine, TEACOP270, is a potent AHR ligand, we were able to show that kynurenine causes the formation of the AHR/ARNT/DRE gel shift complex in vitro (Zheng et al. (2016) Prot. Exp. Pur. 122, 72-81), supporting that kynurenine is an AHR ligand. However, we cannot rule out the possibility that our commercially purchased kynurenine contains TEACOP270.  

  1. There is inadequate interpretation of data for figure 1B, C – line 96 – “...showing that a 2-fold increase of the AHR content caused significant increase of the AHR function.” – Figure 1C displays synergistic activity of CQ and BaP at CYP1A1 mRNA level. On the other hand, Figure 1B shows that combination of CQ and BaP does not affect AhR protein level at all. It is equal to negative control, DMSO. A 2-fold increase is observed for CQ only, but without significant impact on CYP1A1 expression. Thus, there is an impact on AhR activity (function) but not protein level.

Response: The comparison should really be between BaP and CQ/BaP, which shows that 2-fold increase of AHR protein levels causes much more than 2-fold increase of AHR transcriptional activity. Line 85 has been revised in red to provide more clarity to this comparison.

  1. Line 101 – number “6“ from 6AN is a part of a subheadings, 6AN is not introduced for the very first time.

Response: 6AN is now introduced as 6-aminonicotinamide in the abstract (line 20). It is no longer a part of a subheading.

  1. I suggest for most sentences containing a word “treatment” to use the preposition “with”, i.e. Treatment with MG132/6AN/CQ..” – instead of “Treatment of MG132/6AN/CQ,..”.  – lines 153.

Response: We have now changed “treatment of” to “treatment with” accordingly.

  1. Description of figures describing WB data often says: “The images represent the biological triplicates of one experiment.” – However, I believe, this is not correct. If one experiment equals to one passage of cells, then presented data are technical triplicate of one experiment. The authors just divided cellular lysate from one passage. This is technical replicate. I biological one would be from different passage of the cells.

Response: When we did an experiment with biological triplicates, we seeded the cells to three wells of a 6-well plate to equal confluence before starting any treatment. Since the cells were exposed to the same treatment in three different wells, the cell lysates from these wells should be considered as biological triplicates, even though they were from the same passage. Technical triplicate in this case would be loading three western lanes from the same lysate. 

  1. Line 251 – since all experiments were performed in human A549 cell line and the authors detected CYP1A1 mRNA (Figure 1C), it is difficult to understand why there were used rat cells containing rat AhR which may have at least different affinity next to human AhR (which is well described phenomenon).

Response: We agree that it would be ideal to use human cells carrying the DRE-driven GFP cDNA for this study. Our experience with these rat stable cells is that all known AHR ligands we tested show fluorescence. Although there are clear differences of ligand affinities between rat and human AHR, we compared the fluorescence of the treated cells with cells treated with the DMSO control – which is no ligand. We were unable to observe any differences between the DMSO-treated and the 6AN, MG132, or GA treated cells, suggesting that they are not ligand.  

  1. For better orientation, Figure 7C should be aligned similarly to transcripts (Fig 7B) – E-cadherin, vimentin, N-cadherin. Moreover, transcripts after 6AN treatment would be great to demonstrate as well. In Figure 7D – names of the proteins at the right should be closer to data, since e.g.Vimentin seems to belong to middle picture.

Response: Fig. 7C, D, E have now been aligned to be in this order: E cadherin, N-cadherin, and vimentin, like in Fig. 7B. Names of the protein have now been aligned closer to the figures. We agree that including the transcript levels of 6AN-treated cells would make the data look even, but we argue that the protein levels of these markers are more important and interesting than the transcript levels.

  1. At line 244 it is stated: “Results from the time-dependent experiment showed that high dose of GA suppressed the AHR protein levels more effectively than low dose of GA (Fig. 5E), supporting that high dose of GA promotes AHR degradation via autophagy. However, we cannot rule out the possibility that higher dose of GA may also cause an increase of the proteasomal degradation of AHR.”  – Since there are no data that would demonstrate that higher dose of GA is a better autophagy inducer in A549 cell line, authors should either reformulate the sentences or detect e.g. LC3-I, II after GA treatment or ubiquitin on AhR as a proof of GA-induced proteasomal degradation (e.g. by Co-IP).

Response: We agree. The sentence has now been revised in red as: “…supporting that high dose of GA causes more AHR degradation, possibly through autophagy.” (line 202).

  1. Interpretation at Line 240 “We observed that CQ increased the AHR levels from 29 to 58% whereas MG132 decreased from 38% to 31% (Fig. 5C and D), showing that autophagy, but not proteasomal degradation, is essential for maintaining the AHR protein levels” should be corrected. A minimal difference without significance actually means, these values are equal. If authors want to keep such interpretation than at least 3 other measurements should be performed to get significance.

Response: We agree. The sentence has now been revised in red as: “We observed that CQ increased the AHR levels significantly from 29 to 58% whereas MG132 did not alter the AHR levels in a statistically significant manner (Fig. 5C and D), showing that autophagy, but not proteasomal degradation, is essential for maintaining the AHR protein levels.”

  1. Interpretation at line 242 “The slight reduction of AHR levels by MG132 in this case is probably due to the activation of autophagy by MG132, as observed in Fig. 2D and E.” – is not correct. Figures 2D/2E do not show MG132-triggered autophagy, they demonstrate impact of MG132 on AhR content. Authors should demonstrate at least conversion of LC3-I to LC3-II for MG132 to be sure that MG132 induced autophagy in A549 cells.

Response: Fig. 2E shows that CQ rescues the MG132 effect when we compare MG132 and MG132 + CQ, supporting that the MG132-induced AHR reduction is caused by autophagy. For clarity, we have revised the sentence in red where we reported the Fig. 2C results (line 101).

  1. The interpretation at lines 206-208 “The AHR protein levels were similar from 6h up to 24h after BaP treatment and similar inhibition of AHR autophagy by CQ was observed within that period (1.5 to 1.8-fold), suggesting that autophagy was involved for maintaining the AHR levels after ligand treatment (Fig. 4A-E)” is confusing. It is not clear to what sample(s) the folds are related to. Moreover, schematic figure should be present above WB representative image as the whole figure is difficult to understand.

Response: We agree. We added the fold changes in Fig. 4A-D to show where we got 1.5 to 1.8-fold. The Western images were shown separately as one Western results in Fig. 4E to allow comparison among BaP, BaP/CQ, and BaP/MG132 of all time points. This sentence has now been revised in red (lines 170-176) to provide clarity as follow: “The AHR protein levels were similar from 6h up to 24h after BaP treatment (Fig. 2B, right). Interestingly, similar inhibition of AHR autophagy by CQ was observed within the 6h to 24h period since 1.5 to 1.8-fold increase of the AHR levels was observed when we compared BaP and BaP/CQ treatment groups (Fig. 4A-D). Fig. 4E was the representative Western images of Fig. 4A-D, which showed that from 6h to 24h, the AHR levels were increased in the presence of CQ whereas MG132 did not have any effect. Collectively, we concluded that autophagy is likely involved for maintaining the AHR levels after ligand treatment.”

  1. The description of the data at line 194 is difficult to understand: “Cotreatment of BaP with 100 M CQ, however, increased the AHR protein levels to 1.8-fold…” - compared to what? - This is not specified.

Response: It was compared to cells treated with BaP alone. This sentence has now been revised in red to provide clarity (line 162).

Reviewer 2 Report

Review  ijms-2635475

The manuscript is written well. The Introduction is sufficient, the Authors used many methods to confirm the hypothesis. The manuscript could be accepted to publish when the Authors respond for the questions below:

MINOR

1)     Why the Authors used so high concentration of CQ (60 and 100 uM)?  

2)     Which was the final concentration of DMSO in the control cells?

3)     Please add the information about the level of knockdown of Atg5 and LAMP2

4)     Please add the information about the reference protein used for normalization of the protein levels in western blot. Why the Authors did not show the bands of the reference protein?

Author Response

Responses to Reviewer 2 comments (Reviewer’s comments is in italic):

The manuscript is written well. The Introduction is sufficient, the Authors used many methods to confirm the hypothesis. The manuscript could be accepted to publish when the Authors respond for the questions below:

  1. Why Authors used so high concentration of CQ (60 and 100 uM)?  

Response: We had tried as low as 20uM CQ in human cells to block AHR autophagy and it appears that 40uM CQ is effective for some human cells (such as HeLa) but not others (such as MDA-MB-468 and HaCaT). We believe that the CQ concentration is cell line specific. As for A549 cells, CQ appears to block AHR autophagy experimentally starting from 60uM.

  1. Which was the final concentration of DMSO in the control cells?

Response: We used 1,000X stock for all experiments. Max is to add two different stocks in DMSO per condition. Thus, the final concentration of DMSO is mostly 0.1% and sometimes 0.2%.

  1. Please add the information about the level of knockdown of Atg5 and LAMP2.

Response: 30% and 28% remaining content of LAMP2 and ATG5, respectively, in the corresponding stable knockdown cells. This information has been added in red to the Fig. 3 figure legend.

  1. Please add the information about the reference protein used for normalization of the protein levels in western blot. Why authors did not show the bands of the reference protein?

Response: We use total protein to normalize Western bands. Please refer to response under #4 of Reviewer 1 for future details (please read below).

  1. No loading controls (e.g. actin, tubulin, GAPDH) are present despite whole manuscript is based on western blotting technique.

Response: Correct. We intentionally did not use any reference protein such as beta-actin or GAPDH for normalization because it is more accurate to use total protein of the lane for normalization. This method of normalization is optimal with the use of LICOR imager to determine the intensity of the bands of interest using near-IR secondary antibody. Line 736 under 4.3 states that all western bands are normalized by total protein stain.

Round 2

Reviewer 1 Report

The manuscript entitled:  Activation of chaperone-mediated autophagy inhibits the aryl  hydrocarbon receptor function by degrading this receptor in  human lung epithelial carcinoma A549 cells“ investigates connection between chaperone-mediated autophagy (CMA) and arylhydrocarbon receptor (AhR) level, potentially activity.

In the last review version, most of the raised concerns have been corrected.

But, in this version (pdf file), some remained.

1)      Figure 1A is almost missing – probably formatting issue

2)      The part about recommended nomenclature of CYPs has not been fully understood. It is well accepted, that human genes or cDNAs should be written in upper case letters and italics (e.g. CYP1A1), while mostly in lower case letters for mouse (e.g. Cyp1a1). Despite often used word transcript, actually cDNA was measured by PCR. Since manuscript deals with human cells, upper case version would be more appropriate.

3)      Line 106 – number “6“ from 6AN is still a part of a subheadings 2.1.2

4)      Line 258 - Interpretation at line 258 “ The slight reduction of AHR levels by MG132 in this case is probably due to the activation of autophagy by MG132, as observed in Fig. 2D and E.” – is still incorrect. Figures 2D/2E do no show MG132-triggered autophagy, they demonstrate impact of MG132 on AhR content. This is something I agree with the authors’ response to my comment. However, subordinate clause “as observed in Fig. 2D and E.” refers to MG132-triggered autophagy, which is not demonstrated in figure 2D/E nor C. The figures demonstrate impact of these compounds on AhR protein level only.  Thus, a proper sentence would be “…due to the activation of autophagy by MG132, as demonstrated elsewhere (citation)”.

5)      The response of authors to my previous comment: „Subfigures of a given figure presented in the manuscript should be merged into one figure to obtain better lucidity.“ was „It is challenging to group the data into one figure without compromising clarity. After much thought, we believe that the current form is still the best.“

Despite I fully support the creativity, smashed subfigures of one figure over 2-3 pages, make the manuscript less organized. Since majority of subfigures contain actually two identical set of data (one visual as WB, one statistical of visual), certain level of compression (into 1 figure for 1 page) would still be appropriate.

Author Response

Responses to Reviewer 1 2nd round comments (Reviewer’s comments in italic):

In the last review version, most of the raised concerns have been corrected. But, in this version (pdf file), some remained.

1)      Figure 1A is almost missing – probably formatting issue

Response: My latest download Word doc has Fig. 1A in good form. Probably the reviewer has a different version of Microsoft Word. I trust that the editing process will correct that if there is still a problem.

2)      The part about recommended nomenclature of CYPs has not been fully understood. It is well accepted, that human genes or cDNAs should be written in upper case letters and italics (e.g. CYP1A1), while mostly in lower case letters for mouse (e.g. Cyp1a1). Despite often used word transcript, actually cDNA was measured by PCR. Since manuscript deals with human cells, upper case version would be more appropriate.

Response: All gene names are now revised in red accordingly. The revised gene names (e.g. CYP1A1, AHR) in figures are not in red because it will be hard to change in TIFF format. The real-time RT-PCR measures the cDNA of the transcript. Thus, the word transcript remains unchanged.

3)      Line 106 – number “6“ from 6AN is still a part of a subheadings 2.1.2

Response: I believe the revised subheading 2.1.2 is line 90 now, and I do not see any problem with the word “6AN”. I wonder if it is a problem with a different version of Word doc. I hope that the final editing process will correct that if there is a problem.

4)      Line 258 - Interpretation at line 258 “ The slight reduction of AHR levels by MG132 in this case is probably due to the activation of autophagy by MG132, as observed in Fig. 2D and E.” – is still incorrect. Figures 2D/2E do not show MG132-triggered autophagy, they demonstrate impact of MG132 on AhR content. This is something I agree with the authors’ response to my comment. However, subordinate clause “as observed in Fig. 2D and E.” refers to MG132-triggered autophagy, which is not demonstrated in figure 2D/E nor C. The figures demonstrate impact of these compounds on AhR protein level only.  Thus, a proper sentence would be “…due to the activation of autophagy by MG132, as demonstrated elsewhere (citation)”.

Response: This phrase has now been revised in red accordingly as “…as demonstrated elsewhere [30, 31].” (line 204).

5)      The response of authors to my previous comment: „Subfigures of a given figure presented in the manuscript should be merged into one figure to obtain better lucidity.“ was „It is challenging to group the data into one figure without compromising clarity. After much thought, we believe that the current form is still the best.“ Despite I fully support the creativity, smashed subfigures of one figure over 2-3 pages, make the manuscript less organized. Since majority of subfigures contain actually two identical set of data (one visual as WB, one statistical of visual), certain level of compression (into 1 figure for 1 page) would still be appropriate.

Response: We have now arranged the appropriate subfigures into one TIFF image, namely Fig. 2C-E, 4A-E, 5A-B, and 5C-D, so that it should be easier to compare among them. Although controls in some figures are the same, such as Fig. 2C-D, 2F-G, and 5C-D, these controls should be used to evaluate the tested condition in the same experiment to make sense statistically. Combining the controls would not make sense statistically. As for the controls in Fig. 4A-B, 4C-D, and 5A-B, these controls are different since they were harvested at different timepoints. Showing these figures separately makes sense statistically.